# Rational Design of Ni-Doped V_2_O_5_@3D Ni Core/Shell Composites for High-Voltage and High-Rate Aqueous Zinc-Ion Batteries

**DOI:** 10.3390/ma17010215

**Published:** 2023-12-30

**Authors:** Songhe Zheng, Jianping Chen, Ting Wu, Ruimin Li, Xiaoli Zhao, Yajun Pang, Zhenghui Pan

**Affiliations:** 1School of Materials Science and Engineering, Tongji University, Shanghai 201804, China; zhengsh_crane@163.com (S.Z.); 2010448@tongji.edu.cn (J.C.); 2331542@tongji.edu.cn (T.W.); liruimin0187@link.tyut.edu.cn (R.L.); zhenghuipan@tongji.edu.cn (Z.P.); 2College of Chemistry and Materials Engineering, Zhejiang A&F University, Hangzhou 311300, China

**Keywords:** ZIBs, V_2_O_5_, 3D Ni nanonets, ion incorporation, voltage window, reaction kinetics, cycle stability

## Abstract

Aqueous zinc-ion batteries (ZIBs) have significant potential for large energy storage systems because of their high energy density, cost-effectiveness and environmental friendliness. However, the limited voltage window, poor reaction kinetics and structural instability of cathode materials are current bottlenecks which contain the further development of ZIBs. In this work, we rationally design a Ni-doped V_2_O_5_@3D Ni core/shell composite on a carbon cloth electrode (Ni-V_2_O_5_@3D Ni@CC) by growing Ni-V_2_O_5_ on free-standing 3D Ni metal nanonets for high-voltage and high-capacity ZIBs. Impressively, embedded Ni doping increases the interlayer spacing of V_2_O_5_, extending the working voltage and improving the zinc-ion (Zn_30_^2+^) reaction kinetics of the cathode materials; at the same time, the 3D structure, with its high specific surface area and superior electronic conductivity, aids in fast Zn_30_^2+^ transport. Consequently, the as-designed Ni-V_2_O_5_@3D Ni@CC cathodes can operate within a wide voltage window from 0.3 to 1.8 V vs. Zn_30_/Zn_30_^2+^ and deliver a high capacity of 270 mAh g^−1^ (~1050 mAh cm^−3^) at a high current density of 0.8 A g^−1^. In addition, reversible Zn^2+^ (de)incorporation reaction mechanisms in the Ni-V_2_O_5_@3D Ni@CC cathodes are investigated through multiple characterization methods (SEM, TEM, XRD, XPS, etc.). As a result, we achieved significant progress toward practical applications of ZIBs.

## 1. Introduction

Fossil fuels such as coal and oil play a crucial role in human activities, but rapid reductions in these energy reserves and the environmental damage caused by their usage led to an urgent energy revolution. As a result, fossil fuels are gradually being replaced by clean energy in the form of electricity [1]. In the research field of electric energy storage systems, lithium-ion batteries (LIBs) have been developed for over 30 years [2,3], been the subject of in-depth research and achieved great success in commercial applications [4,5]. However, energy density limitations [6], high cost and environmental issues limit further applications of LIBs [7]. To overcome these issues, researchers are exploring the feasibility of other active metals in energy storage devices to replace LIBs in the future [8,9].

Among all candidates, aqueous zinc-ion batteries (ZIBs) have received increasing attention as promising candidates for large-scale energy storage devices [10] due to the apparent advantages of using metallic Zn as the anode, including its high theoretical capacity (5855 mAh cm^−3^) [11], low redox potential (−0.76 V vs. a standard hydrogen electrode (SHE)) and low-cost benefit from its vast reserve (the cost of Zn (0.5–1.5 USD/lb) is much smaller than that of Li (8–11 USD/lb)) [12,13,14]. However, finding suitable cathode materials for high-performance ZIBs has become a top priority [15,16,17,18,19]. Currently, manganese- [20], vanadium- and Prussian blue analogue (PBA)-based cathodes have been proven to be typical cathode materials for ZIBs [21]. Among them, manganese- [22,23] and PBA-based [24] materials have ideal charging and discharging platforms [25] but usually suffer from an insufficient long-term cycling ability [26] and low reversible capacity [27] because of their intrinsic unstable structural characteristics [28,29,30]. Therefore, vanadium-based materials with a high theoretical specific capacity and adjustable ion-transfer channels will most likely become promising candidates for high-performance cathodes in ZIBs [31].

Generally, the narrow interlayer spacing and poor electronic conductivity of vanadium-based materials can lead to slow reaction kinetics and unstable structures during the Zn^2+^ (de)incorporation process, limiting their applications in ZIBs [32]. For this issue, the incorporation of metal ions, such as Na^+^, Li^+^, Mg^2+^, Mn^2+^, and Cu^2+^ [33,34,35,36,37], has been classified as one of the most effective strategies to enlarge interlayer spacing. In addition, metallic element incorporation adjusts the composition of V-based materials while changing their structures and thus improves electronic conductivity [38]. Therefore, this incorporation strategy significantly improves the materials’ reaction kinetics by hastening both ion transportation and electron transportation. However, the insertion of metal ions may reduce the theoretical capacity of vanadium-based materials, especially during long-term cycle performance, due to irreversible structural destruction, leading to a reduction in vanadium ion storage. Therefore, the influence of the type and amount of pre-embedded metal ions should be considered when introducing metal ions into vanadium-based materials. In addition, most reported cathode materials are in a powdered form. In order to apply these materials as cathodes, the powder must be stacked and bonded on a current collector, which not only blocks the active sites but also increases the “dead mass”, decreasing the energy density of ZIBs and causing inevitable waste. In this regard, loading electrode active materials on 3D frameworks has been proven to be an effective strategy to avoid the use of adhesives and the accumulation of active materials due to their large specific surface area and high porosity. Developing a solution to address both of these crucial problems simultaneously would be highly effective yet challenging [39].

Herein, we propose a rational design of Ni-doped V_2_O_5_@3D Ni core/shell composites on carbon cloth (Ni-V_2_O_5_@3D Ni@CC) for high-performance ZIBs. Specifically, 3D metallic Ni nanonets were grown on the surface of carbon cloth (CC). Compared with CC, Ni nanonets can be applied as a 3D framework to provide high electronic conductivity and a larger surface area for achieving a higher quantity and the full usage of active material loading. Additionally, during the one-step V_2_O_5_-loading process, metallic Ni particles were released from 3D nanonets and incorporated into the V_2_O_5_, modifying the interlayer structure of the active materials. As a result of this innovation, the as-obtained Ni-V_2_O_5_@3D Ni@CC electrode can work in a wide voltage window of 0.3 to 1.8 V versus Zn/Zn^2+^ and deliver a high capacity of 270 mAh g^−1^ (~1050 mAh cm^−3^) at a current density of 0.8 mA g^−1^. Moreover, multiple characterization methods were applied to investigate the reversible Zn^2+^ (de)-incorporation reaction mechanism. Our strategy aims to broaden the application range to other materials and drive the practical commercial usage of ZIBs.

## 2. Experimental Section

Chemicals. Carbon cloth (1 cm × 2 cm), hexamethylenetetramine (HMT, C_6_H_12_N_4_, 99%, Sinopharm Chemical Reagent Company Limited, Beijing, China), nickel chloride hexahydrate (NiCl_2_·6H_2_O, 99%, Sigma-Aldrich, Shanghai, China), hydrogen peroxide solution (H_2_O_2_, 30%, Jnan Mama Technology Company Limited, Wuxi, China), zinc sulfate (ZnSO_4_·7H_2_O, 99.5%, Sinopharm Chemical Reagent Company Limited, Beijing, China), and vanadium pentoxide (V_2_O_5_, Xiya Reagent, Chengdu, China) were used directly without any further purification.

Fabrication of Ni(OH)_2_@CC: Firstly, the purchased CC was soaked in a 5 M HCl aqueous solution to experience a complete cleaning; it was then washed in deionized (DI) water and pure ethanol, each for 15 min in turn, to remove extra HCl. After that, the CC was dried in air. The clean CC was then cut into 1 cm × 2 cm pieces which were used as substrates for the surface deposition of Ni(OH)_2_. Clean CC was protected by ethoxylate except for an exposed area of 1 cm^2^ on both sides. The Ni precursor solution was obtained by adding 0.125 mol of NiCl_2_·6H_2_O and 0.25 mol of HMT into 1 L of DI water and completely mixing the solution. The previously mentioned CC substrate with an exposed area of 1 cm^2^ was immersed in 30 mL of a Ni precursor solution in a 50 mL glass beaker. The beaker was heated in an electric oven at 100 °C for 8 h to synthesize Ni(OH)_2_ on the CC. During this process, light green products were generally grown on the surface of the CC. Finally, after being completely rinsed with distilled water and pure ethanol in turn several times and dried in a vacuum oven for over 12 h, the Ni(OH)_2_@CC samples were successfully fabricated.

Preparation of 3D Ni@CC: To obtain 3D Ni@CC, the fabricated Ni(OH)_2_@CC samples were placed into a porcelain boat and then heated under an Ar/H_2_ mixed atmosphere (the volume percent of H_2_ was 5%) at 400 °C for 1 h. The temperature increase rate was set at 5 °C min^−1^. During this process, Ni(OH)_2_ was transformed into Ni. After cooling the samples to room temperature, 3D Ni@CC samples were obtained.

Preparation of Ni-V_2_O_5_@3D Ni@CC: A Ni-V_2_O_5_@3D Ni@CC electrode was prepared using a simple hydrothermal synthesis method. First, 0.91 g of divanadium pentaoxide was added into 40 mL of hydrogen peroxide solution. Subsequently, the above solution and the 3D Ni@CC were transferred into a 100 mL Teflon-lined stainless-steel autoclave. After being sealed, these materials were heated to 180 °C and maintained for 24 h. After cooling the materials to room temperature, the autoclave was removed, and the obtained samples were fully cleaned using ethanol; they were then placed into an oven to dry overnight at 80 °C. The samples were then annealed in air at 300 °C for 1 h at a heating rate of 2 °C min^−1^ to obtain a Ni-V_2_O_5_@3D Ni@CC. The mass loading of Ni-V_2_O_5_ was about 2.0 mg cm^−2^.

Electrochemical measurements: All electrochemical performances of the samples were measured using a CHI 660C electrochemical workstation (CH Instruments Inc.). The Ni-V_2_O_5_@3D Ni@CC cathode’s performance in full-cell tests was investigated using CR2025 coin cells. We chose a glass microfiber filter (Whatman, Maidstone, British) as a separator and bare Zn foil (which was washed with ethanol) as the anode material. A 2 M ZnSO_4_ electrolyte was obtained by dissolving enough zinc sulfate in DI water. The processes of assembling and testing the coin cells were all carried out in an open atmosphere at room temperature.

Characterization: A Quanta 400 FEG scanning electron microscope (SEM) and a Tecnai G2 F20 S-Twin transmission electron microscope (TEM) (FEI, Hillsboro, State of Oregon, USA)were used to characterize the microstructures of these samples. The SEM was operated from 5.0 to 20.0 kV, and the TEM was operated at an accelerating voltage of 200 kV. The crystal structures of the samples were analyzed by X-ray diffraction (XRD) with a Bruker D8 diffractometer (Bruker, Karlsruhe, Germany). The results of the X-ray photoelectron spectroscopy (XPS) characterization were shown using an Axis Ultra DLD X-ray photoelectron spectroscope (Shimadzu, Tokyo, Japan), and all data were referenced to C 1s = 284.8 eV.

## 3. Results and Discussion

The preparation process of the Ni-V_2_O_5_@3D Ni@CC electrode is illustrated in Figure 1a (the details can also be found in the Experimental Section), while scanning electron microscope (SEM) images obtained during different steps can be seen in Figure 1b–g. CC has excellent characteristics, including high electrical conductivity, remarkable structural flexibility and robust mechanical strength, which allowed for its direct use as the current collector in this work. In addition, metal oxides/hydroxides can be grown easily on CC, thus avoiding the use of an extra binder in the electrode fabrication process. But CC still has an intrinsic disadvantage which is its small surface area, resulting in a low mass loading of active materials [40]. Therefore, to overcome this disadvantage, in the first step, Ni(OH)_2_ nanosheets were grown on the CC surface. The SEM images of the Ni(OH)_2_@CC can be seen in Figure 2b,c in which the nanosheets were vertically grown on a bare CC through immersion in the solution and the reaction between HMT and NiCl_2_·6H_2_O. We can see from the enlarged image that the large number of flexible Ni(OH)_2_ nanosheets provided a much larger surface area in comparison with the CC. The 3D metallic Ni nanonets (Figure 1a) were then derived from the Ni(OH)_2_ nanosheets via a reduction treatment under a H_2_ atmosphere. It can be seen that the obtained 3D metallic Ni nanonets still maintain a large surface area, and the morphology of the Ni(OH)_2_ transformed from nanosheets into well-dispersed and rod-shaped 3D nanosheets offered a more sufficient space for active materials to load (Figure 1d,e). This 3D Ni@CC can also serve as an electrically conductive and stable support for an active material which, in this study, is V_2_O_5_. Additionally, Ni can act as incorporation ion to intercalate active material, changing the internal structure, facilitating ion and electron transportation and making the oxidation–reduction reaction occur quickly. As for the Ni-pollution problem, it mainly refers to the environmental damage caused by the Ni metal smelting process. But in this research, the raw material used for the synthesis of Ni metal is environmentally friendly Ni(OH)_2_, which is usually a product obtained from the alkaline treatment of industrial wastewater containing Ni. Afterward, nickel exists in its elemental form in the cathode of the ZIB which is stable and easy to recycle, greatly reducing its negative impact on the environment.

Then, after a simple hydrothermal synthesis method, V_2_O_5_ was grown on the surfaces of the 3D Ni nanonets. The addition ratios of raw materials underwent tests from low to high to obtain the most suitable ratio between 3D Ni nanonets and Ni-V_2_O_5_ to maximize the usage of surface area for a better electrochemical result. For example, the additive amount of NiCl_2_·6H_2_O was changed from 0.100 mol to 0.200 mol. As the amount continued increase, the final performance did not improve and even deteriorated. We believe that this can be attributed to the continued growth of Ni(OH)_2_, which finally led to heavier 3D Ni nanonets with smaller specific surface areas which were and easier to break down during the V_2_O_5_ loading process, which damages the electrochemical performance of cathodes. Finally, we concluded that the ideal additive amount of NiCl_2_·6H_2_O is 0.125 mol.

The SEM image of the as-synthesized Ni-V_2_O_5_@3D Ni@CC demonstrates that the V_2_O_5_ nanosheets were anchored homogeneously on the surfaces of the 3D Ni nanonets to form unique core–shell composites (Figure 1f,g). It should be noticed that during the V_2_O_5_ loading process, Ni atoms were released from Ni nanonets, achieving incorporation into the interlayered V_2_O_5_. The Ni-V_2_O_5_@3D Ni@CC results in a higher energy density and transfer rate of the ZIB’s electrode while maintaining high stability. More in-depth characterization of this part will be shown later.

We carried out high-angle annular dark-field (HAADF) scanning transmission electron microscopy (STEM) to further investigate the nanostructures of the Ni(OH)_2_@CC, 3D Ni@CC and Ni-V_2_O_5_@3D Ni@CC samples (Figure 2a–c). From Figure 2a, we can see that abundant flexible Ni(OH)_2_ nanosheets were synthesized well on the CC. After a reduction process, the Ni(OH)_2_ nanosheets were transformed into 3D metal Ni nanonets on the surface of the CC with a loose rod-shaped structure (Figure 2b). Finally, Ni-V_2_O_5_ nanosheets were grown on the Ni nanonets through a hydrothermal process. Through a higher-resolution STEM image and elemental characterization of the Ni-V_2_O_5_@3D Ni@CC (Figure 2c,d), not only can robust V_2_O_5_ be seen but the incorporation of the Ni element can be proved as well. There is an element distribution characterization of the sample shown in Figure 2d, in which the bright parts represent areas with physical objects of the Ni-V_2_O_5_ nanosheet. We selected a very small area from Figure 2d by using a red dashed box called a spectrum image to analyze its construction. The green pixels represent the V element, and the blue pixels represent the Ni element. It can be easily understood that due to the large mass loading of the 3D Ni nanonets fully covered with V_2_O_5_, the V element is uniformly distributed on the sample, which is reflected in the image as uniformly distributed green pixels. Blue pixels are also uniformly distributed on the image instead of concentrated on the area of the Ni nanonets, showing that Ni achieved good incorporation with V_2_O_5_ and was uniformly distributed in it. In addition, electron energy loss spectroscopy (EELS) results from this area further demonstrate the occurrence of V_2_O_5_ growth and Ni incorporation (Figure 2e). EELS utilizes an incident electron beam to create inelastic scattering in a sample, and the energy loss of the electrons directly reflects the mechanism of scattering, the chemical composition of the sample and the sample’s thickness information. Therefore, it can analyze the elemental composition and other information from the analyzed area. The blue vertical lines in Figure 2e display different energy losses caused by different elements in which two characteristic peaks belonging to V and Ni elements were detected which coordinate with the result from Figure 2d.

Apart from the SEM and TEM images, our research also characterized the components and structures of samples through X-ray diffraction (XRD) and X-ray photoelectron spectroscopy (XPS). As shown in Figure 3a and Appendix A, the XRD patterns of different samples can be indexed well to Ni(OH)_2_ (#38-0715), metallic Ni (#04-0850) and V_2_O_5_ (#77-2418), indicating that all these samples were fabricated well, as expected. According to the XRD standard card of V_2_O_5_ (#77-2418), we chose three characteristic peaks (20.3°, 31.1°, 34.3°) to give an accurate description of the peaks shifting. In comparison with those characteristic peaks of the pure V_2_O_5_@CC sample (20.3°, 31.1°, 34.3°), the corresponding peaks of the Ni-V_2_O_5_@3D Ni@CC (19.8°, 30.7°, 34.0°) shifted toward lower degrees; according to the Bragg equation, 2*d* sin*θ* = *λ* (*d* represents the distance between adjacent crystal layers, *θ* represents the angle between the incident X-ray and the crystal plane and *λ* is the wavelength of the X-ray), if the characteristic peak shifts toward a small angle while *λ* remains constant, d will be enlarged, representing a bigger interlayer space. It can be determined through the Bragg equation that the d-spaces were enlarged by about 0.5 Å. This evidence strongly proves that the interlayer spacing of the samples is enlarged due to the incorporation of Ni ions. In addition, high-resolution XPS spectra of the Ni element from the 3D Ni@CC and Ni-V_2_O_5_@3D Ni@CC are also presented in Figure 3b, further demonstrating the process of Ni ion incorporation. Clearly, in comparison with the 3D Ni@CC sample, the Ni^0^ of the Ni element in the metallic nanonet sample disappeared, and the main valence state became Ni^2+^ in the Ni-V_2_O_5_@3D Ni@CC (Figure 3b). In addition, according to the XPS spectra for V_2_O_5_ only vs. Ni-incorporated V_2_O_5_, there are shifts in the peak energies for vanadium and oxygen (Appendix A). Since Ni is incorporated within the interlayer spacing, it can be deduced that the intercalation of Ni not only changed the inner structure but also affected the electronic environment of V and O, causing shifts in the binding energies. In total, this evidence indicates the occurrence of Ni incorporation, which is highly consistent with the results from the energy-dispersive spectroscopy (EDS) elemental mappings obtained using the SEM (Appendix A). It can be speculated that during the hydrothermal process of V_2_O_5_ loading, Ni^0^ was released from the Ni nanonets and transferred into Ni^2+^ to incorporate with V_2_O_5_, intercalating into its layered structure to achieve Ni-V_2_O_5_ and then growing on 3D nanonets. Therefore, 3D Ni nanonets could not only serve as frameworks for the improved growth of V_2_O_5_ nanosheets but could also participate in adjusting the V_2_O_5_ layered structure. In other words, both the active site and spacing were enhanced for the diffusion of Zn^2+^ through such a facile one-step strategy.

The ZIBs were assembled by using metallic Zn as the anode to couple with the Ni-V_2_O_5_@3D Ni@CC cathode or the V_2_O_5_@CC cathode (denoted as Ni-V_2_O_5_@3D Ni@CC-based ZIB or V_2_O_5_@CC-based ZIB) and 2 M ZnSO_4_ as an electrolyte. The corresponding electrochemical properties were also analyzed. The oxygen evolution reaction (OER) activity is an important parameter for describing cathode stability in ZIBs. A higher reaction potential means that the electrode material has better stability during cycle performance. Meanwhile, a cathode material with a higher OER potential can provide a wider voltage window for ZIBs to realize better performance closer to commercial usage. As can be seen from Appendix A, during charging, there is a higher potential (~1.9 V) required for the Ni-V_2_O_5_@3D Ni@CC electrode to start decomposition and release oxygen as the current density begins to increase rapidly, while the V_2_O_5_@CC only allows for a lower potential at about 1.6 V.

We then produced cyclic voltammetry (CV) curves at a scan rate of 1.0 mV s^−1^. In comparison with a pure V_2_O_5_@CC-based ZIB (0.3–1.5 V), the Ni-V_2_O_5_@3D Ni@CC-based ZIB realizes a wider working potential window of 0.3–1.8 V; this can be attributed to the Ni-V_2_O_5_@3D Ni@CC electrode with a lower OER activity, as we mentioned before. The wider voltage window enables the ZIB to work at a high energy density (Figure 4a and Appendix A). In addition, we also note that this working potential window is larger than that of other recently reported V_2_O_5_-based cathodes (e.g., Zn_0.25_V_2_O_5_, Ca_0.25_V_2_O_5_ and Na_0.33_V_2_O_5_) [36,41,42,43,44,45].

Galvanostatic charge–discharge (GCD) curves of the V_2_O_5_@CC-based ZIB and Ni-V_2_O_5_@3D Ni@CC-based ZIB at a current density of 0.8 A g^−1^ were then created to compare the specific capacity (Figure 4b). Obviously, the Ni-V_2_O_5_@3D Ni@CC electrode delivers a much higher capacity than that of the V_2_O_5_@CC. Additionally, as for the Ni-V_2_O_5_@3D Ni@CC electrode, there is an obvious charge/discharge platform at about 1.2 V which is obviously higher than that of the V_2_O_5_@CC cathode. This is another powerful piece of evidence proving that the improved cathode material is more suitable for practical usage not only because of its high energy density and stability but also because of its wider working voltage window and higher working voltage platform, which are all better than those of the V_2_O_5_@CC electrode. Figure 4d shows the GCD curves of the Ni-V_2_O_5_@3D Ni@CC under various current densities (from 0.4 to 4.8 A g^−1^), from which obvious charge and discharge plateaus can be discerned, consistent with the CV result (Figure 4c). During these two tests, it can be observed that the working voltage platform was maintained steadily, ignoring various changes in current density, proving that the Ni-V_2_O_5_@3D Ni@CC electrode is an ideal cathode material for ZIBs used in different circumstances. It should be noticed that even though the voltage window appears to grow smaller as the current density increases, this can be explained because the difference in CV curves is caused by capacity. The capacity of a cathode material always decreases under a faster voltage scan rate since Zn ions cannot be (de)intercalated completely. Therefore, the CV curve area grows small while the voltage window is remaining the same.

In the rate performance tests, the Ni-V_2_O_5_@3D Ni@CC-based ZIB presents average discharge capacities of 330, 270, 216, 166, 152 and 140 mAh g^−1^ at current densities of 0.4, 0.8, 1.6, 3.2, 4.0 and 4.8 A g^−1^, respectively. Moreover, when the current density returns to 0.8 A g^−1^, the discharge capacity recovers to 270 mAh g^−1^ (~1050 mAh cm^−3^), suggesting an excellent rate performance. Meanwhile, the Ni-V_2_O_5_@3D Ni@CC exhibits a high Coulombic efficiency (CE) at different current densities ranging from 0.4 to 4.8 A g^−1^ and then back to 0.8 A g^−1^ (Figure 4e). In a long-cycle stability test, even under a high current density of 4.8 A g^−1^, the Ni-V_2_O_5_@3D Ni@CC-based ZIB still maintains a nearly non-decreased 100% CE after cycling over 500 times with a specific capacity of about 110 mAh g^−1^ (a volume specific capacity of about 430 mAh cm^−3^, shown in Figure 4f). The performance reported here is better than that of other V_2_O_5_-based compounds such as V_2_O_5_/CC, V_2_O_5_/MXene, NH_4_^+^-V_2_O_5_ and V_2_O_5_/rGO [46,47,48,49,50]. Meanwhile, the smaller resistance of the Ni-V_2_O_5_@3D Ni@CC-based ZIB in comparison with the V_2_O_5_@CC-based ZIB, proved by an electrochemical impedance spectroscopy (EIS) test (which can be seen in Appendix A), demonstrates well that the incorporation of Ni into the V_2_O_5_ active material not only benefits the ZIB’s energy storage performance and cycle stability but also improves the Zn^2+^ transport ability inside the cathode, giving the Ni-V_2_O_5_@3D Ni@CC-based ZIB better application potential. This can be explained by the fact that the incorporation of Ni ions not only enlarged the interlayer space of the V_2_O_5_, allowing more Zn ions to intercalate, but also broadened Zn^2+^ ion transport channels, hastening their moving speed inside the electrode active material. Meanwhile, the broadened channels can escape from structural breakage occurring during Zn^2+^ ion transportation and then improve the stability of the cathode during cycling.

In order to gain more insight into the (de)incorporation process of Zn^2+^ in the Ni-V_2_O_5_@3D Ni@CC-based ZIB, an ex situ XRD analysis was performed. As displayed in Figure 5, during the discharging process, a characteristic peak of the Ni-V_2_O_5_@3D Ni@CC electrode appeared at the 8.32° position, representing the appearance of an interlayer structure due to the intercalation of Zn^2+^ ions inside the cathode. The characteristic peak shifted toward a lower angle (8.08° at 0.6 V) as the potential decreased until the fully discharged state (7.92° at 0.3 V) was reached. During the charging process afterwards, the characteristic peak shifted back to higher angles (8.04° at 0.7 V and 8.08° at 1.1 V) and then disappeared again, just like the beginning, after being fully charged to 1.8 V. During the discharge process, Zn^2+^ ions are gradually intercalated into the cathode, resulting in a contraction of interlayer spacing which can be further recovered upon voltage reversal. This is strong evidence proving that the energy storage mechanism of the Ni-V_2_O_5_@3D Ni@CC electrode is Zn^2+^ ion (de)intercalation. Meanwhile, the small shift in the characteristic peaks represents a smaller interlayer–structure change during the charge/discharge process due to the existence of Ni ions, leading to a more stable cathode material.

## 4. Conclusions

In summary, we designed Ni-doped V_2_O_5_@3D Ni core/shell composites on CC as an advanced cathode material for high-performance ZIBs. Three-dimensional Ni nanonets with electronic conductivity composited on CC can provide a high specific surface area for V_2_O_5_ loading which can overcome the inevitable disadvantage of CC. During the V_2_O_5_ loading process, through a hydrothermal reaction, Ni atoms are transferred from Ni^0^ to Ni^2+^ and subsequently released from Ni nanonets to incorporate with V_2_O_5_. The intercalated Ni ions enlarged the interlayer space of the loaded V_2_O_5_, which is favorable for Zn^2+^ intercalation and transportation while maintaining its intrinsic structure. The as-prepared Ni-V_2_O_5_@3D Ni@CC electrode delivered a wide voltage window of 0.3 to 1.8 V versus Zn/Zn^2+^ and presented a high capacity of 270 mAh g^−1^ at a current density of 0.8 mA g^−1^. Moreover, even under a higher current density of 4.8 A g^−1^, the Ni-V_2_O_5_@3D Ni@CC-based ZIB still maintained almost 100% CE after cycling over 500 times. Our research provides a new pathway to realize better performance for V-based cathode materials and makes significant progress toward the practical applications of commercial ZIBs. In further research, we can allocate more resources to systematically study the usage of alternative materials and their impact on battery performance. This will allow us to obtain research results with reduced costs, simplified preparation methods and increased practical value. The “3D metallic structure and incorporation” strategy can be expected to extend to other metals and electrode active materials and cooperate with other improvement strategies, including electrolyte and anode research areas in ZIBs, which will also favor the further design of high-performance ZIBs.

## Figures and Tables

**Figure 1 materials-17-00215-f001:**
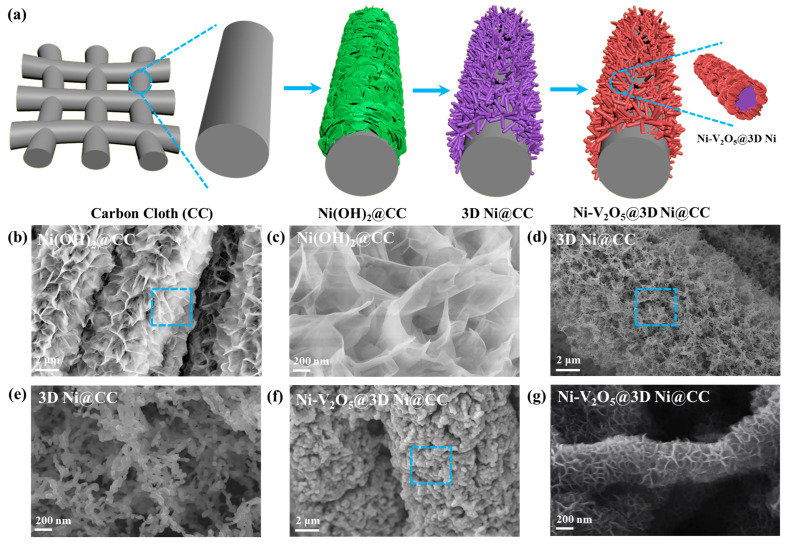
(**a**) Illustration of the preparation process of the Ni-V_2_O_5_@3D Ni@CC electrode. (**b**–**g**) SEM images of (**b**,**c**) Ni(OH)_2_ nanosheets synthesized on carbon cloth (CC); (**d**,**e**) 3D porous Ni nanonets on CC (3D Ni@CC) and (**f**,**g**) Ni-V_2_O_5_@3D Ni@CC. All the latter images in each pair are partial enlargement of the blue squares in the previous images.

**Figure 2 materials-17-00215-f002:**
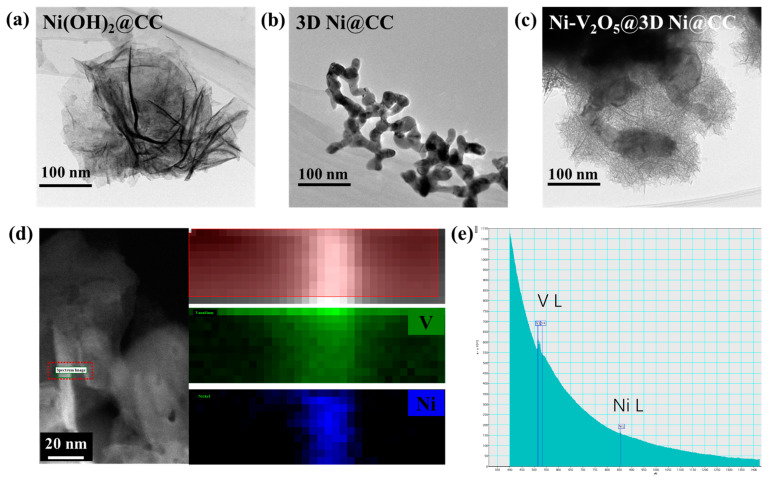
TEM images with higher magnification of (**a**) Ni(OH)_2_@CC, (**b**) 3D Ni@CC and (**c**) Ni-V_2_O_5_@3D Ni@CC. (**d**) STEM image and element distribution of a Ni-V_2_O_5_ nanosheet. (**e**) EELS result of a Ni-V_2_O_5_ nanosheet taken from the red dashed box in (**d**).

**Figure 3 materials-17-00215-f003:**
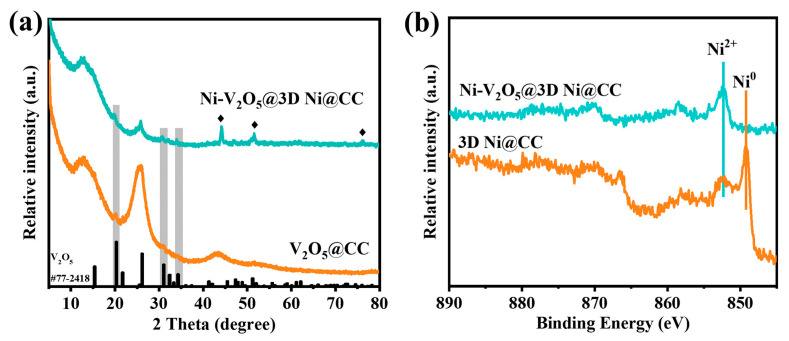
(**a**) XRD patterns of V_2_O_5_@CC and Ni-V_2_O_5_@3D Ni@CC. (**b**) XPS spectra of 3D Ni@CC and Ni-V_2_O_5_@3D Ni@CC.

**Figure 4 materials-17-00215-f004:**
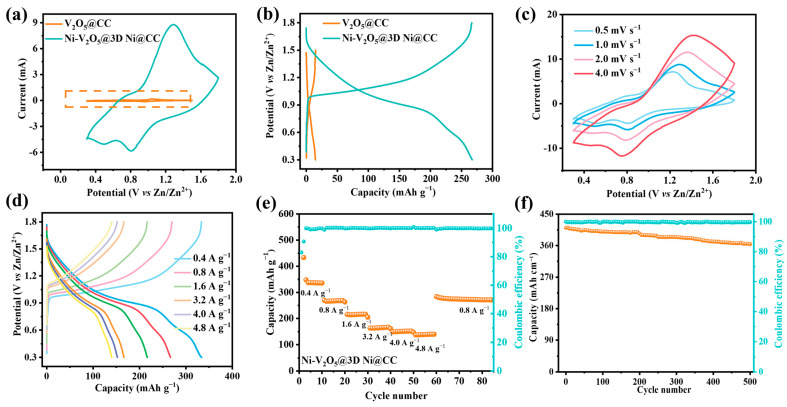
(**a**) CV curves of V_2_O_5_@CC-based ZIB (0.3–1.5 V) and Ni-V_2_O_5_@3D Ni@CC-based ZIB (0.3–1.8 V) at a scan rate of 1.0 mV s^−1^. (**b**) Galvanostatic charge–discharge (GCD) curves of V_2_O_5_@CC-based ZIB and Ni-V_2_O_5_@3D Ni@CC-based ZIB under a current density of 0.8 A g^−1^. (**c**) CV curves of Ni-V_2_O_5_@3D Ni@CC-based ZIB at different scan rates from 0.5 to 4.0 mV s^−1^. (**d**) GCD curves of Ni-V_2_O_5_@3D Ni@CC-based ZIB under different current densities (0.4–4.8 A g^−1^). (**e**) Rate capability test of Ni-V_2_O_5_@3D Ni@CC-based ZIB under various current densities (0.4–4.8 A g^−1^). (**f**) Long-cycle stability test for Ni-V_2_O_5_@3D Ni@CC-based ZIB at a current density of 4.8 A g^−1^.

**Figure 5 materials-17-00215-f005:**
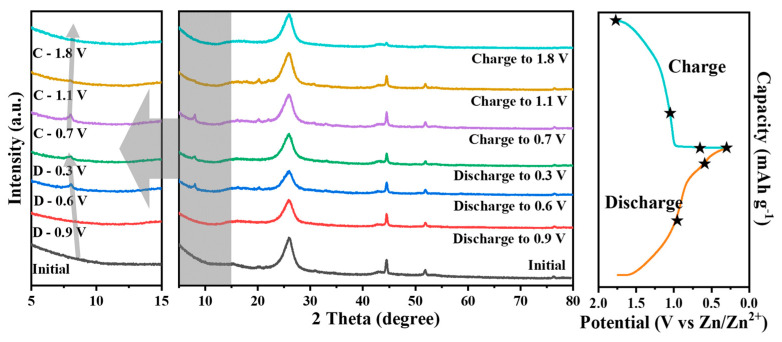
Ex situ XRD patterns for Ni-V_2_O_5_@3D Ni@CC electrode during the charge/discharge process under different potentials at a current density of 0.4 A g^−1^. The potential corresponding to each curve in the XRD image are represented by stars in the right charge/discharge curve.

## Data Availability

Data are contained within the article and Appendix A.

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
