# Peer review of "Rational Design of Ni-Doped V2O5@3D Ni Core/Shell Composites for High-Voltage and High-Rate Aqueous Zinc-Ion Batteries"

_materials, 2023, doi:10.3390/ma17010215_

Round 1
Reviewer 1 Report
Comments and Suggestions for Authors
The authors here describe a modification of V2O5 as cathode material for zinc ion batteries based on Ni-incorporated V2O5 built on 3D Ni nanonets supported on carbon cloth electrode which report wider voltage window of 0.3 to 1.8V and higher capacity of 270 mAh g-1. While the article does show new findings related to the improved capacity with the Ni incorporation, the characterization of the electrode material seems lacking and needs further characterization to prove the structure that the authors are hypothesizing.
1) The authors have EELS spectra collected for the material, however the data presented is very vague and the analysis is not very clear. Can the authors more descriptive of what is being looked at here? The image from EELS doesn’t seem to match the STEM image that is shared. Also, what does the pink color represent?
2) I don’t understand what I should be looking at Fig. 2e. Can the authors explain more on what they are trying to show here? Also, the image is not very clear to the reader.
3) In the XRD spectra, the authors report shift in the peak towards lower degrees. Can the authors be more specific on what peaks and by what amount is the shift? How repeatable is this data across different prepared samples?
4) For the XPS peaks, can the authors also add XPS spectra for V2O5 only vs Ni incorporated V2O5? Do we have peaks for V and O detected in the XPS spectra post V2O5 deposition (would recommend collecting spectra before the deposition as control)? Is there a shift in the peak energies for vanadium/oxygen? If Ni is incorporated within the interlayer spacing, I would expect the incorporation of Ni to also affect the electronic environment of V, causing a shift in the binding energies.
5) Can the authors also do a SEM-EDX on the samples for Ni(OH)2, 3D-Ni, and post V2O5 deposition to determine the elemental composition for all the elements (V, O, Ni, C)? This would basically prove the elemental composition of the material the authors are proposing here.
6) In the ex-situ spectra, the authors again report a shift towards higher angles in fully discharged state of the characteristic XRD peaks. Can the authors pin-point to explain what peaks they are referring to and what is the shift?
7) The references in reference section don’t have proper citation format. Most of the references don’t have the journal name mentioned.
8) Page 2, Line 61, there is a typo. Should be “powder” instead of “power”.
9) Formatting for Figure 4 is very poor. Please increase the font size on the axis labels for each of the figures
Comments on the Quality of English LanguageMinor editing required, especially in the introduction section
Reviewer 2 Report
Comments and Suggestions for Authors
The manuscript ID materials-2768624 mainly presents a study about the synthesis and standard characterization of a particular Ni-doped V2O5@3D Ni core/shell composite on carbon cloth electrode. The design envisioned potential applications for aqueous zinc ion batteries. Please see below a list of comments to the authors:
1. It is not clear how were selected some of the parameters in the synthesis in order to guarantee that the study is systematic instead of incidental.
2. Please comment about the reproducibility of the data plotted in figure 4.
3. The content of Ni can make nonlinear changes in electrical effects. The authors are invited to discuss and see for instance: http://dx.doi.org/10.1016/j.mseb.2017.03.004
4. The advantages and disadvantages of the material proposed should be highlighted in the discussion section in order to easily see the value of the work. You can see comparative alternatives, for instance: https://doi.org/10.1016/j.ensm.2020.02.016
5. A graphical abstract describing the methodology and main results of the study would be helpful for readers.
6. How is the avoided the incorporation of oxygen during the incorporation of Ni in the samples?
7. Some perspectives should be described for future work based on the main findings.
8. The main results must be confronted with updated publications of the topic of zinc ion batteries.
9. Some of the citations presented in collective form could be split in order to better justify the selection of the importance of the references for this topic by individual expressions.
10. Error bar in experimental data should be incorporated.
Comments on the Quality of English LanguageA proofreading is mandatory
Reviewer 3 Report
Comments and Suggestions for Authors
General Comments
The paper provides valuable insights into the development of a promising cathode material for ZIBs. However, improvements are necessary in providing specific details regarding characterization techniques, cost evaluation, and economic feasibility for commercial utilization. Additionally, ensure consistency in acronyms and accurate referencing throughout the manuscript.
Consider addressing these points to enhance the manuscript's clarity, precision, and relevance for publication.
Abstract
The abstract presents a clear overview of the paper's objectives and findings regarding the development and evaluation of a novel cathode material for Zinc-ion batteries (ZIBs). However, it would benefit from minor improvements:
- Include atomic numbers as subscripts and specify the state of charge for Zn (superscript) to enhance clarity and precision.
- The statement, "investigated by multiple characterization methods," lacks specificity. Please provide details within parentheses regarding the characterization techniques utilized.
Introduction
The introduction sets the context effectively but requires some clarification and citation updates:
- Refinement is needed in line 34-35 regarding the statement about reduced costs. Please cite specific reports or studies supporting the claim. Additionally, elaborate on the environmental concerns related to materials like cobalt and nickel.
- Maintain consistency in using acronyms throughout the manuscript. Consider uniformly employing acronyms for terms such as "manganese" and "vanadium."
- Correct punctuation in references; ensure consistency in formatting by removing the dot after references such as [45-47], and [39, 48-50].
Lines 64, 66, and 79 require clarification on the characterization techniques used and should be specified explicitly. Furthermore, line 81 would benefit from additional elaboration regarding the practical implications and commercial viability of the findings. Include details on cost assessments and economic feasibility, particularly in scaling up the novel cathode material design for commercial use.
Experimental Section
The Experimental Section lacks sufficient detail in the characterization subsection. Specify the setup configurations of the instruments used for characterization to enhance reproducibility and understanding of the methods employed.
Comments on the Quality of English LanguageThe author's use of the English language meets the publication standards, albeit with minor room for improvement. Specifically, consolidating some of the shorter sentences would enhance the overall flow and coherence of the discourse.
Reviewer 4 Report
Comments and Suggestions for Authors
The manuscript reported a fabrication of Ni-doped V2O5/Ni/carbon cloth hybrid for zinc-ion batteries. Writing was concise with solid evidence. The reviewer recommend the manuscript to Materials for publishing after the authors appropriately solve those concerns:
1/ Those numbers/questions should be declared:
-Mass loading of Ni-doped V2O5/Ni on carbon cloth
-How to calculate the specific capacity? Based on mass of Ni-doped V2O5 or the whole electrode mass?
-(page 5) which planes were enlarged? How big were the new d-spaces?
2/ The authors mentioned in the Introduction that the theoretical capacity of Zn anode, 5855 mAh cm-3, made ZIBs attractive. Because the authors mentioned the unit of capacity as mAh cm-3, the authors should also use that unit for their electrode characterization. At least one graph, in which the unit of capacity is mAh cm-3, should be introduced.
Round 2
Reviewer 1 Report
Comments and Suggestions for Authors
The authors have addressed most of my concerns. The manuscript is now acceptable for publication.
Comments on the Quality of English LanguageMinor editing is required. Would recommend running through English language editing before publication
Author Response
We really appreciate the reviewer for the agreement for manscript publication. For the requirement of English language editing, we have carefully reviewed the entire manuscript and made revisions to all grammar and wording mistakes within it. Hoping it can be satisfied for publication.
Reviewer 2 Report
Comments and Suggestions for Authors
The authors have clarified all the points raised in the review stage. The results are interesting and the conclusions about nanocomposites for ion batteries are solid. Then I can recommend this work for publication in present form.
Comments on the Quality of English LanguageA proofreading is suggested
Author Response
We really appreciate the reviewer for the agreement for manscript publication. For the requirement of proofreading, we have carefully reviewed the entire manuscript and made revisions to all grammar and wording mistakes within it. Hoping it can be satisfied for publication.
Reviewer 4 Report
Comments and Suggestions for Authors
2. The authors refused using the unit "mAh cm-3" in presenting their results, but used for the introduction. The authors said that the advantages of ZIBs over LIBs was the high volumetric capacity of Zn metal. So that advantages will (and probably should) disappear if anode is not Zn metal. In case of using Zn metal anode, why should the evaluation of cathode is in gravimetric but the one of anode is in volumetric? Anode alone cannot make the full ZIBs cell become more compact. If later research wants to estimate the volumetric capacity of the full cell based on material only, knowing volumetric capacity of cathode is compulsory.
The authors must introduce at least one graph presenting the volumetric capacity of the research target (in unit of mAh cm-3).
Author Response
We thank the reviewer for the thorough evaluation of this manuscript, and appreciate the comment to make the manuscript more valuable and meaningful. We have already edited the manuscript by simultaneous use both mass & volume specific capacity, and one graph presenting the capacity of cathode materials has also been conversed in unit of mAh cm-3 in Fig. 4f, so that later research can estimate the volumetric capacity of the full cell based on material only.
Hoping this manuscript can be satisfied for pubilication.